# Using evidence when planning for trial recruitment: An international perspective from time-poor trialists

**Heidi R. Gardner** *, **Shaun Treweek, Katie Gillies**

Health Services Research Unit, Health Sciences Building, University of Aberdeen, Aberdeen, Scotland, United Kingdom

* heidi.gardner1@abdn.ac.uk

## Abstract

### Introduction

Recruiting participants to trials is challenging. To date, research has focussed on improving recruitment once the trial is underway, rather than planning strategies to support it, e.g. developing trial information leaflets together with people like those to be recruited. We explored whether people involved with participant recruitment have explicit planning strategies; if so, how these are developed, and if not, what prevents effective planning.

### Methods

Design: Individual qualitative semi-structured interviews. Data were analysed using a Framework approach, and themes linked through comparison of data within and across stakeholder groups.

Participants: 23 international trialists (UK, Canada, South Africa, Italy, the Netherlands); 11 self-identifying as 'Designers'; those who design recruitment methods, and 12 self-identifying as 'Recruiters'; those who recruit participants. Interviewees' had recruitment experience spanning diverse interventions and clinical areas.

Setting: Primary, secondary and tertiary-care sites involved in trials, academic institutions, and contract research organisations supporting pharmaceutical companies.

### Results

To varying degrees, respondents had prospective strategies for recruitment. These were seldom based on rigorous evidence.

When describing their recruitment planning experiences, interviewees identified a range of influences that they believe impacted success:

- The timing of recruitment strategy development relative to the trial start date, and who is responsible for recruitment planning.

- The methods used to develop trialists' recruitment strategy design and implementation skills, and when these skills are gained (i.e. before the trial or throughout).

**Data Availability Statement:** All relevant data are within the paper and its Supporting Information files.

**Funding:** This study was funded by the Chief Scientist Office of Scotland's Health Improvement,

Protection and Services Research Committee (project reference HIPS/16/07 - https://www.cso.scot.nhs.uk/outputs/cso-funded-research/hips16/). HRG was supported by a scholarship from Aberdeen Development Trust which funded her PhD fees and stipend, KG was supported by an MRC Methodology Research Fellowship (MR/L01193X/1), and ST was supported by core funding from the University of Aberdeen. The funders had no role in study design, data collection and analysis, decision to publish, or preparation of the manuscript.

**Competing interests:** The authors have declared that no competing interests exist.

- The perceived barriers and facilitators to successful recruitment planning; and how trialists modify practice when recruitment is poor.

## Conclusions

Respondents from all countries considered limited time and disproportionate approvals processes as major challenges to recruitment planning. Poor planning is a mistake that trialists live with throughout the trial. The experiences of our participants suggest that effective recruitment requires strategies to increase the time for trial planning, as well as access to easily implementable evidence-based strategies.

## Introduction

Patient participants are central to the success of randomised controlled trials. The implications of poor participant recruitment threaten the completion of trials, and where poorly-recruiting trials do stagger on without being prematurely closed down then poor recruitment threatens the utility of their results [1]. Researchers have therefore tried to improve the evidence-base with regard to identification of effective recruitment interventions. Currently there are limited numbers of robustly evaluated successful strategies, few of which are generalisable to a wide range of trial populations and settings [2]. With no evidence-base to inform recruitment planning, it is not surprising that trialists around the world struggle to ensure that their trials successfully recruit to target [2,3].

Exploratory research aiming to understand recruitment difficulties has largely centred on perceptions and experiences of participants that have taken part in trials, or those individuals who have declined to take part [4–9]. Whilst these studies are useful when looking at barriers and facilitators for prospective participants, research into the trialists' views and experiences of recruitment is needed to ensure that methodologists are able to design trials that are acceptable for both participants and trialists. With this in mind, more recent research has focussed on getting to grips with what the process of recruitment is like for trial teams [10–16]. Several studies have highlighted the process of recruitment from the perspective of the recruiter [11]; suggesting that many recruiters find it difficult to combine research alongside their clinical roles [12–16, 17], and that this combination of responsibilities can result in significant emotional challenges that may arise from the conflicting priorities of research and clinical care [15]. These studies have been able to provide a more complete picture of the recruitment process and have highlighted the importance of recruitment barriers at various levels. Building a research-friendly culture within healthcare systems is known to impact recruiters as individuals [10], whereas significant system-level barriers arise resulting from conducting research within the healthcare system [10].

This study provides an additional layer of knowledge to the evidence that has explored trial recruitment. Investigating what is happening in trials that are ongoing has been the focus of most of the work to date [10, 17, 18]. However, there is merit in taking a step back and exploring how trialists make decisions that lead to the design and implementation of recruitment processes; the process of recruitment planning. This study employed a semi-structured qualitative interview approach in order to explore how trialists plan for participant recruitment; a key part of that is identifying whether or not evidence is used by trialists when planning their

recruitment strategies, as well as exploring the perceived barriers and facilitators to the use of this evidence. We also wanted to explore if and how trialists' experiences contrasted with the perceived level of planning required by funders and approvals bodies before a trial begins, and how recruitment techniques may or may not change when participants are not being recruited at forecasted levels.

## Methods

### Ethics statement

This study was approved by the University of Aberdeen College of Life Sciences and Medicine College Ethics Review Board (CERB/2016/6/1382). Signed informed consent was obtained from all participants, which included consent for anonymised quotes from their interviews to be published in publications and presentations resulting from this work.

### Study design

Our qualitative interview study included two groups of participants; Recruiters (frontline staff involved in contacting and identifying potential participants and working to actively recruit participants; such as Research Nurses, GPs and Clinicians), and Designers (the people who are able to influence the strategies used for recruitment such as Trial Managers, Principal Investigators and Chief Investigators).

### Setting

We focussed on Phase 3 pragmatic effectiveness trials conducted within primary, secondary or tertiary care settings, whilst also working to include trials conducted within the community, and trials funded by industry that often take place at purpose-built sites (e.g. contract research organisation sites that are used only for research).

Participants were invited through a range of networks: Trial Forge collaborators (at the time of the study this encompassed 12 Trials Units, two of which were outside the UK), the MRC Trial Methodology Hubs, the UK CRC Trials Units, the UK Clinical Research Networks, the UK Trial Managers' Network, and other relevant networks (e.g. the Association of Clinical Research Professionals and the Institute of Clinical Research). Prospective participants were asked to respond by email to express interest. Interested participants were sent written study information, including an information leaflet and a consent form, and were provided with an opportunity to ask questions before making their decision. Participants were given the choice of face-to-face, telephone or Skype interview.

### Sample size

We aimed to recruit a minimum of 20 participants. Research has shown that with relatively consistent participant groups, data saturation can occur within the first twelve interviews; although overarching themes and concepts can be present as early as six interviews in [19,20]. In the trials field it is relatively common for Designers to have experience of roles associated with the Recruiter category from earlier on in their career, we therefore anticipated some level of consistency between the two groups. A formal assessment of whether adequate thematic saturation had occurred was not employed [21].

### Sampling

We purposively selected participants based on their trial portfolios; aiming to include a diverse range of funders (e.g. private, public, third sector), clinical specialities, and intervention type

(e.g. investigational medicinal product, licensed drug, surgical technique, medical device, and behavioural interventions (lifestyle change)).

## Research team and data collection

A conversational session within Centre for Healthcare Randomised Trials' monthly Trials Group meeting at the University of Aberdeen guided the development of topic guides. We invited 12 participants to the meeting, and involved a mix of stakeholder groups from within the Unit; Trial Managers, Data Coordinators and Programmers were all represented. HRG chaired the session and facilitated discussion on the general theme of recruitment planning, and what participants wanted to know about how other trial teams plan their recruitment.

The topic guide (Supporting information S1 and S2 Files) developed was then employed in individual semi-structured interviews conducted by HRG. At the start of each interview, participants were encouraged to discuss their practical experiences of recruiting participants to trials, and their perspective on the recruitment process. Interviews were conversational yet supported by the topic guide to ensure that key issues were covered. The topic guide was refined throughout the study, and field notes were taken after each interview to assist analysis and interpretation.

## Analysis

Data were analysed using the Framework method; a qualitative approach that suited this study due to its specific research questions, pre-designed sample, a-priori issues and deducible themes [22]. The Framework method has been used widely and successfully for applied health services research [10, 23–26].

All interviews were audio recorded, transcribed verbatim and anonymised before analysis. HRG coded the first four transcripts using an open coding approach to develop a working analytical framework. KG independently reviewed a 10% sample of transcripts. Coding and themes were discussed by the team (HRG, KG and ST) to agree the analytical framework that would be applied to all interviews. We elected to categorise themes into areas of the trial timeline. HRG applied the analytical framework to all transcripts and used NVivo to generate a framework matrix to facilitate comparison of data first within, and then across stakeholder groups (Recruiters and Designers). Following analysis, we selected relevant quotes representing pertinent themes to illustrate study findings. As per our ethics statement, all participant quotes presented here have been anonymised to protect confidentiality.

## Results

### Participants

Twenty-five trialists from the Recruiter and Designer stakeholder groups were invited to secure 23 interviews which lasted between 32 and 77 minutes (median: 58 minutes).

Key characteristics of the study sample are presented in Table 1.

When asked to describe their experiences with trial recruitment planning, interviewees were able to identify a range of influences which they considered as having played a role in either positively or negatively impacting the success of trial recruitment. As the following data will show, the points at which planning impacted on the success (or not) of trial recruitment interventions and/or strategies was varied. Participants from both Recruiter and Designer stakeholder groups discussed an assortment of influences throughout the recruitment pathway and beyond, reflecting their diverse professional backgrounds and trial experiences. Principal Investigators and Chief Investigators had a mix of experience; many had previous clinical experience, and others had backgrounds in nursing, midwifery and allied health professions.

**Table 1. Characteristics of interviewees.**

| Interviewee characteristics | Recruiter | Designer |
|---|---|---|
| Stakeholder group | 11 | 12 |
| **Location** | | |
| UK | 7 | 11 |
| Canada | 0 | 1 |
| The Netherlands | 1 | 0 |
| Italy | 1 | 0 |
| South Africa | 2 | 0 |
| **Gender** | | |
| Male | 1 | 3 |
| Female | 10 | 9 |
| **Age (years)** | | |
| 30 and under | 6 | 1 |
| 31–50 | 4 | 6 |
| 51 and above | 1 | 5 |
| **Experience of working in clinical trials** | | |
| Less than 10 years | 7 | 4 |
| 10 years or more | 4 | 8 |
| **Trials background**[*] | | |
| Public | 8 | 8 |
| Private | 5 | 3 |
| Third sector | 1 | 1 |
| **Involvement with clinical research networks and speciality groups**[±] | | |
| Scottish Primary Care Clinical Research Network | 3 | 2 |
| Scottish Cancer Clinical Research Network | 2 | 2 |
| Scottish Stroke Clinical Research Network | 4 | 3 |
| English Clinical Research Network | 1 | 5 |
| Scottish Musculoskeletal Speciality Group | 0 | 1 |
| UKCRC Registered Clinical Trials Unit Network | 8 | 7 |

Note: [*] and [±] provide brief information on the types of trials and clinical research networks that participants have experience with. A number of participants had experience in more than one of these sub-categories (e.g. both public and private trials, and/or contact with the Scottish Primary Care Clinical Research Network, Scottish Cancer Clinical Research Network, and English Clinical Research Network), hence why figures may total to more than the number of participants in each round of user testing.

They encompass: pre-trial recruitment planning, to what extent recruitment strategies and interventions should be planned in advance of the commencement of participant recruitment, and who is responsible for recruitment planning; the methods and/or resources that trialists use to develop their recruitment intervention/strategy design and implementation skills and when those skills are developed (i.e. before a trial begins, or learning as the trial is ongoing); perceived barriers and facilitators to successful recruitment planning; and how trialists modify their practice when recruitment is lower than anticipated.

## Responsibility for recruitment planning

Both Recruiters and Designers valued the importance of planning when it came to recruitment strategies. Unsurprisingly, Designers perceived this planning to be a task within their job

specification, and one that they understood was a key component to successful recruitment once the trial had started. On the whole, they perceived recruitment planning to be a useful investment of time, but clearly felt that time was something of a scarcity.

"*When you start off in research you are told you should spend two thirds of your time thinking about the project and one third of your time doing it, but you don't bother with that. You just get started*! *Spend 5% on preparation, then spend a huge amount of time undoing all the mistakes that you've made*" (Professor and Research Director–Designer, UK, Participant 6).

## Detailed planning of recruitment methods during trial design stages

The time invested in planning and working-up a trial before funding is awarded was a significant point of discussion across both stakeholder groups.

When asked to consider recruitment planning in terms of the detail required during the process of grant writing, both Designers and Recruiters perceived that specific details about planned methods for recruitment were not necessary for grant applications. The need for detailed planning of the operationalisation of how potential participants will be recruited was thought to be increasing when applying for trial grants; this was an issue referenced several times by participants with experience in applying to UK-based funders.

"*So I haven't seen whole sections of grant applications that are devoted to that* [recruitment], *although I think some of the funders like* [UK funding body], *and others, are starting to indicate that information about that is required. So, I think that the requirements of researchers have not been particularly stringent when it comes to specifying feasibility and recruitment, and likely recruitment potential.*" (Clinician–Recruiter, UK, Participant 4).

"*Different funders ask for different amounts* [of detail]. *It's gradually getting more that they ask for.*" (Principal Investigator and Clinician–Designer, UK, Participant 2).

However, a couple of experienced Designers perceived this to be a task that should be done in advance of grant application submission regardless of whether it was a requirement for funding or not, chiefly because it may increase chances of funding success;

"*I advise that they do write all those* [planned recruitment methods] *into the application, so it really gives the funders confidence that we've really thought about it clearly.*" (Trial Manager–Designer, UK, Participant 3).

Once grant applications have been submitted and funds awarded, funders were perceived to take a hands-off approach to recruitment, only providing direct involvement if targets were not being met.

"*Once we have a plan in place, they will simply monitor that we're achieving our targets.*" (Principal Investigator and Clinician–Designer, UK, Participant 2).

All interview participants agreed that at least some information on how to recruit participants should be included within the protocol, but with regards to level of detail there was a diverse range of experiences among participants. Recruiters tended to perceive the required level of detail as limited, and that the Trial Manager and the research team running the study would produce a practical plan for these processes based on the fundamental detail held within the protocol. In other words, a document separate from the protocol on 'how to' recruit.

*"I would say not always very much* [detail]. *It's pretty much you know, we will recruit by... I think in our protocol it might have said,* "We're going to use these two methods [referring to two methods discussed previously; mail outs from GP, and putting posters up in clinical involved with the trial]". *But the actual finer point of how that's going to happen isn't listed ... it's up to the Trial Manager and the research team to come up with that."* (Specialist Research Nurse–Recruiter, UK, Participant 5).

Designers however, suggested that writing the protocol for recruitment activity required more foresight.

*"So a couple of times we've had to do amendments to the protocol to incorporate new recruitment strategies. So, I mean I now, having had several years' experience, then I advise that they do write all those into the grant application so it really gives the funders confidence that we've really thought about it clearly and that there's you know, we've got contingency plans."* (Trial Manager–Designer, UK, Participant 3).

Designers with significant trial experience (10 years or more) explained how they purposefully build imprecision into the protocol to circumvent the need for future ethical amendments. Interestingly, the imprecision that they build in to their protocols was all related to projected recruitment figures rather than the development of recruitment strategies.

*"You don't really want to put the target for randomisation is 3,000 patients, because if you go to 3,001 patients some regulators will ask you to put in an amendment and go back to ethics. So, we would always write,* "Randomise at least 3,000 patients" *or,* "Around 3,000 patients in at least 100 centres". *So, you put in imprecision in your protocol which we discovered that most ethics committees and sponsors and other regulators don't notice."* (Principal Investigator and Clinician–Designer, UK, Participant 2).

## Use of empirical and experiential evidence in recruitment planning

Only one participant explicitly mentioned use of the empirical evidence from the Cochrane recruitment review as informing their decisions about the content of recruitment plans.

'*Strategies to improve recruitment to randomised controlled trials*';

*"I look at what's been effective before, so for example, I'm aware of the Cochrane Reviews about trials, and about the evidence about what improves recruitment, so I know that if you provide money to people, they want shorter questionnaires–we've looked at all that evidence and we do try to take that on board."* (Professor and Chief Investigator–Designer, UK, Participant 16).

Commonly, methods for recruitment planning tended to have been fostered and refined as a result of experience from working on multiple trials. Both Recruiters and Designers felt that there were advantages in being aware of what works for others (i.e. experiential evidence) with regard to recruitment methods used in similar trials. Each of the Designers interviewed had developed their own methods of planning for recruitment. Largely these centred on use of experiential evidence from skilled colleagues, as illustrated by the following quote:

*"Basically we do a lot of canvassing of opinion, getting some expert advice* [earlier in the interview participant referred to a number of 'experts' by name–all experienced (10 years or more) trialists]." (Trial Manager–Designer, UK, Participant 8).

When participants were asked about what they would do if recruitment was not going well, experiential evidence was discussed at length. Designers explained that the first step would usually be to connect with colleagues with experience in similar types of trials to assess existing strategies and work together to think of new strategies that may be appropriate;

"*Inevitably when you are faced with a challenging situation about recruiting there tend to be people that you would go to, so for example, someone like... if I was recruiting in Primary Care, someone like* [trialist's name] *would be someone that I would have a word with.*" (Professor and Chief Investigator–Designer, UK, Participant 16).

### Perceived barriers and facilitators to successful recruitment planning

**Effective communication and learning from others.** Effective communication was common theme across both stakeholder groups, with participants highlighting the need for effective communication of experiential evidence specifically;

"*Talk to them about why it's not working because I just think that communication is the only way that you're going to get anywhere.*" (Specialist Research Nurse–Recruiter, UK, Participant 11),

There was also an observation that in the experience of Designers, teams that successfully recruit to target are likely to be more communicative, and as a result more resilient than teams that are less communicative. One experienced Designer explained that;

"*You have to have an explicit communication strategy built into that, appropriate to the... whatever the study is. Some of it is just factual communication, but some of it is more the social thing, making a team cohesive and resilient whenever the problems arise.*" (Principal Investigator and Clinician–Designer, Canada, Participant 9).

**Time as a limiting factor for recruitment planning.** Both Recruiters and Designers perceived the scarcity of time to be a notable issue that could lead to stress or frustration in the run up to the opening of recruiting sites.

"*It can be very stressful, especially having a particular you know, "This is your... you must recruit by this point." And also it depends you know, on some trials there's an awful lot to do before you even get to that point.*" (Specialist Research Nurse–Recruiter, UK, Participant 5).

The time needed to plan and set up a trial is often dictated by funding bodies and the timing between confirmation of funding and the start of the grant; this issue is therefore largely out of the trial teams' control, and can result in staff working against the clock to try and obtain approvals in order to get recruitment started on time. This was illustrated neatly by a participant that said;

"*I think the only other thing that is frustrating is when you do get a grant for a trial or you're putting in for a grant for a trial, there's often not enough time allocated for planning and setting up. So it's always a bit of a mad rush you know.*" (Trial Manager–Designer, UK, Participant 8).

**Inconsistencies with research approvals and governance procedures.** Research approvals and governance was a topic covered by all participants. There was a feeling of frustration with regards to how the approvals process is implemented in the UK; particularly surrounding the level of inconsistency observed when working across multiple sites and/or health boards. Inconsistencies were viewed as a point of significant irritation, something that could impact on the quality of recruitment planning and standardisation of recruitment methods used across sites, but not necessarily something that Recruiters and Designers could confidently say was a reason for poor recruitment. These irregularities in decision-making by regulators and ethics committees tended to involve subtle changes to the wording of documentation or method of approach.

*"I also think that there is still too much inconsistency between committees in how they may make decisions, and there is also a problem at the committee level of them feeling that they always need to adjust patient information leaflets, or consent forms to, as it were, justify their regulatory position."* (Clinician–Recruiter, UK, Participant 4).

## Perceptions of the recruitment process

Throughout the interviews, the majority of participants voiced their opinions of the recruitment process more generally. Initially, their thoughts did not appear to be directly linked to recruitment planning, but upon further investigation these experiences provide an insight into the environment that many trialists are working in. Pressure was a topic covered frequently both by Recruiters and Designers. Largely, Recruiters had experienced, or heard of other colleagues experiencing, *'a lot of pressure'* to recruit. These stakeholders expressed that their efforts were, *"constantly never good enough. Never good enough. It was constantly, "You need to get people in the door" to the point where you had to go and deliver leaflets to social places to try and get people in."* (Specialist Research Nurse–Recruiter, UK, Participant 5), with pressure coming from funders in the majority of cases. Designers viewed this pressure from an alternative perspective; showing empathy for the Recruiters, and a willingness to provide support to them when appropriate.

Although the international sample was small (5/23 participants were based outside of the UK), the experiences referenced by both international and UK-based participants were relatively similar, particularly when it came to the way that recruitment strategies are developed and implemented. The main difference between international and UK-based experiences arose during interviews with trialists working in countries with a substantially different healthcare system. For example, one of the interviewees working in South Africa explained how trials were often easier to recruit to when they were being conducted within the public healthcare system. In these cases patients that are unable to afford private healthcare are more likely to consider trial participation, as they are given the chance to see healthcare professionals more quickly.

## Discussion

This study explored experiences and perceptions of the process of planning recruitment strategies for trials. Our findings provide insight into the way that trialists plan (or not) their recruitment strategies, how these skills are developed over time, and the potential barriers and facilitators that impact on the effectiveness of any planning that is undertaken.

The vast majority of qualitative studies focussed on trial recruitment to date have investigated barriers and facilitators to effective recruitment from the perspective of prospective

participants, and the majority of work looking at recruitment from the trialists' standpoint has centred on parts of the active recruitment process; i.e. when trialists approach potential participants to talk to them about trial participation. Whilst these studies have generated useful data on the recruitment process, so far the literature has lacked exploration of the planning that precedes it.

## Communication between trial stakeholders

All of our study participants believed communication to be an important component of recruitment success. Communication involves layers of conversations and information exchange between various trial stakeholders; where communication falters recruitment is expected to stumble too.

Poor communication is an issue between the trial team, and external stakeholders. Both Recruiters and Designers referenced their frustration at the lack of consistency between approvals and governance procedures across multiple sites; explaining that variations impacted on recruitment planning and standardisation of methods, but they could not confidently say that this caused poor recruitment. These experiences are echoed throughout the literature, though most reports are written by frustrated researchers and represent only single applications [27–39].

## Managing expectations

Open channels of communication have the potential to facilitate productive working environments [40], this extends to the management and communication of realistic expectations. Our findings highlight frustration with unrealistic forecasted recruitment figures, a scenario so common that there is significant literature on the topic. Lasagna's Law [41, 42] and Muench's Third Law [43] state that '*the number of patients available for entering a trial falls markedly at study initiation and rises markedly after study completion*' [42] and '*the number of cases promised in any clinical study must be divided by a factor of at least 10*' [43] respectively. Recent qualitative work focussed on projection of recruitment numbers in primary care provides additional insight into factors that contribute to poor forecasting [44]. Optimistic figures have been attributed to inappropriately 'anchoring' estimates by focussing on positive past experiences [45], failing to consider significant differences between studies, and the 'Lake Wobegon Effect' where individuals overestimate their achievements relative to the average [46]. Unsurprisingly, this pressure impacts members of the Recruiter stakeholder group in particular; they are tasked with speaking to potential participants and recruiting to the trial. Designers often express empathy for Recruiters, but the problem can be avoided if predicted recruitment figures are intentionally less optimistic and therefore more dependable from the beginning of the trial.

## Clarity and culture change

Both Recruiters and Designers considered the content of grant applications when asked about recruitment planning. These discussions centred on the contrast between the level of detail that funders require, and what trialists feel is necessary for them to get to grips with the process of recruitment i.e. the operationalisation of recruitment. Our interviewees explained that overwhelmingly, trial teams focus on the 'what' of recruitment, i.e. how many participants are required for the trial to answer its research question, and over what time period, rather than the 'how', i.e. what strategies should be used, where and when. The experiences of our participants suggest that this level of detail is largely accepted by funders '*as long as it sounds like you* [trial teams] *know what you're doing*'. Funders have a key role here. In the UK, public and

patient involvement (PPI) is now an established part of the funding landscape, with funders requiring detail on how PPI will be used throughout a change. A similar change could be take place for recruitment planning–funders and reviewers should consider operationalisation of the recruitment activity.

## Generation and implementation of recruitment evidence

The process of culture change is slow, but given that the Cochrane recruitment review was originally published in 2007 [47], a lack of awareness or of use of the most relevant source of systematically reviewed evidence around participant recruitment over a decade later, is worrying. Just one participant explicitly referenced use of the Cochrane recruitment review without being prompted to do so, and the financial incentives the individual referred to are actually part of the Cochrane retention review [2], How the results of systematic reviews about trial methods are disseminated needs to be improved because trialists are largely unaware of them at present.

This is perhaps not surprising since at 185 pages long the Cochrane recruitment review (for example) does not present itself as an efficient way of gathering new knowledge quickly. As one interviewee who had heard of the Cochrane recruitment review put it: "[I] *had been meaning to read it"* and had not found time to do so.

Findings from linked work [48] suggests that trialists want to see information presented in a layered format. This information should begin with a simple explanation of the intervention, followed by bite-sized chunks about its impact on recruitment and the level of certainty in the evidence. Once trialists have an outline of the intervention they then want to be able make a decision about accessing further details, rather than being confronted with all of the information to begin with. Our work suggests that the context in which the intervention has been tested (participant population, trial intervention and study location), and what information we still do not have about the intervention (e.g. other contexts, cost, potentially negative implications), are priorities to trialists [48].

## Looking beyond improved recruitment rates

This project aimed to provide researchers with evidence about if and how the recruitment process is planned. We hope that this information, along with a significant body of additional work that has been published, and is in the process of being planned, conducted, and disseminated, can be used to ultimately improve trial recruitment figures. We focus on the issue of low participant recruitment not because we believe it will solve every issue that a clinical trial may have, but because it is an avoidable issue that too many trials encounter. As Whitham and colleagues make clear, there are other things that trialists need to stay on top of for a trial to be a success [49].

The potential for selection/participant bias [50] is a problem that we need to be aware of when considering the participants that we recruit [51], and whether they are retained until the end of the trial. Planning inclusive recruitment processes that provide the general population of patients that the trial's results may apply to with opportunities to participate, is imperative to ensuring that trials provide useful results.

## Strengths and limitations

A significant strength of this study was the high level of diversity in interviewees; Specialist Research Nurses, Research Managers, Investigators, Trial Managers, Clinicians and Clinical Trial Educators from the UK, South Africa, Italy, the Netherlands and Canada explained how they plan for participant recruitment. Our findings reflect experiences of trialists working in

various environments, both with and without a trials unit, and across disease areas; the experiences we found were consistent, suggesting that our findings will apply to trialists outside of the immediate study population.

One of our recruitment methods was to approach people through the existing list of Trial Forge collaborators; this produced an engaged sample, only 2 people approached declined to take part. It is likely that study participants were interested in trial recruitment, potentially signifying existing knowledge of recruitment methods. That said, our findings demonstrate that even those aware of trial methods research and Trial Forge struggle with recruitment and are largely not aware of the existing evidence base.

## Implications for practice

To make the most of time spent planning recruitment strategies, it is important that stakeholder groups communicate effectively. Regulators and other approvals bodies should be working with the research community to ensure that the burden of governance processes do not overshadow the research that they intend to facilitate., Trialists and funders need to communicate to ensure that funders provide sufficient time between confirmation of funding and the start of the trial, to ensure that trialists are able to work with colleagues to produce recruitment methods that are sufficiently thought-through. Potential changes to funding could include a staged approach to funding release for the sole purpose of planning. One idea may be that outline stages for trial grant calls are shorter pitches that, if successful, award smaller funding pots to employ someone to work up operations for the full grant proposal for a period of 3 months or so. Following those three months of dedicated planning, the proposal could be reviewed by the funder, who would release funds for the full-scale trial only when the planning process has been approved.

Less complex operationalised change may come by simply ensuring that trial teams are aware of the existing evidence around recruitment interventions, and that there is an expectation on the part of funders for evidence-based interventions to be used where possible. Initiatives such as Trial Forge (of which this work is a part of) and the UK MRC-NIHR Trials Methodology Research Partnership (https://www.methodologyhubs.mrc.ac.uk/about/tmrp/) are working to strengthen connections with funders to ensure that rigorously evaluated interventions are implemented where possible, and/or that embedded recruitment studies ('SWATs') are introduced into trials that could plug a gap in the existing evidence base [3, 52]. That said, the issues associated with planning recruitment strategies are multifaceted, and require multi-stakeholder collaboration.

## Conclusions

This study has highlighted the complexity of planning trial recruitment strategies. The work involved can be lengthy and is often rushed as a result of time pressure. It is important that trialists, regulators, and funders recognise this process as an essential part of a trial's workload, and that as a community we seek to alleviate the barriers and enhance the facilitators to effective recruitment planning. When trialists experience poor recruitment, they tend to implement multiple strategies based on experiential evidence; meaning that robust empirical evidence is rarely generated. The problem is not that there is nowhere for manuscripts covering these kinds of topics to be published (Trials, PLOS One, BMJ Open, and the Journal of Clinical Epidemiology have all published trials methodology research), the problem is that trialists are time-poor and therefore struggle to find the time to test the methods that they are implementing. Where robust evidence does exist, w must work to ensure that trialists have unobstructed access to, know about and use these rigorously evaluated strategies s. Often these strategies are

shared at relevant conferences such as the biennial International Clinical Trials Methodology Conference (UK-based) and the annual Society of Clinical Trials Meeting, but as these events require in person attendance, dissemination and expertise sharing is limited to those that can attend. Video recording or remote attendance using online conferencing suites would be one way to improve the accessibility of these events, therefore facilitating knowledge exchange and sharing of expertise with trialists around the world.

If and when evidence-based approaches have been exhausted, we need to encourage implementation of embedded studies that effectively generate evidence that is useful to the wider trials community. There is a significant body of literature on survey methodology covering topics such as incentives and questionnaire length/content [53, 54, 55], which may act as a source of inspiration for testing strategies that have not yet been tested in the trial recruitment sphere. In addition, the Northern Ireland Hub for Trials Methodology Research holds the SWAT repository [56], which provides details (including outcome measures and analysis plans) of ongoing SWATs that can be implemented by trial teams. Once multiple trials have tested the same SWAT, the results can then be pooled to generate results that shed light on if and how the intervention operates across trials in a variety of contexts.

Ultimately, we must add structure to the process of recruitment planning; currently trialists rely on experienced colleagues and experiential evidence, which may be useful in the short-term, but does not offer a sustainable route for long-term evidence-sharing. The process of *recruitment* planning ultimately needs to be an integral part of the *trial* planning process and encouraging opportunities for operationalising the recruitment plan will go some way to start that process.

## Supporting information

**S1 File. Topic guide for designers.**
(DOCX)

**S2 File. Topic guide for recruiters.**
(DOCX)

## Acknowledgments

The authors would like to thank all interviewees that took part in this study, as well as all of the trial staff who assisted with identifying potential participants. The authors would also like to thank the trial staff from Aberdeen's Centre for Healthcare Randomised Trials that took part in the initial session which led to the development of the topic guides used throughout this study.

## Author Contributions

**Conceptualization:** Heidi R. Gardner, Katie Gillies.

**Data curation:** Heidi R. Gardner.

**Formal analysis:** Heidi R. Gardner, Katie Gillies.

**Funding acquisition:** Heidi R. Gardner, Shaun Treweek, Katie Gillies.

**Investigation:** Heidi R. Gardner, Katie Gillies.

**Methodology:** Heidi R. Gardner, Shaun Treweek, Katie Gillies.

**Project administration:** Heidi R. Gardner.

**Supervision:** Shaun Treweek, Katie Gillies.

**Writing – original draft:** Heidi R. Gardner.

**Writing – review & editing:** Heidi R. Gardner, Shaun Treweek, Katie Gillies.

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
