## [Decision Letter · Decision Letter 0]

9 Sep 2019

PONE-D-19-17368

Using evidence when planning for trial recruitment: An international perspective from time-poor trial recruiters and designers

PLOS ONE

Dear Dr Gardner,

Thank you for submitting your manuscript to PLOS ONE. After careful consideration, we feel that it has merit but does not fully meet PLOS ONE’s publication criteria as it currently stands. Therefore, we invite you to submit a revised version of the manuscript that addresses the points raised during the review process.

In particular, the reviewers would like further information to justify the study, with some background on recruitment models, and discussion on the potential risk of bias that may occur with recruitment difficulties. Further information on the possible sources of uncertainty in recruitment would also benefit readers.  Please also carefully address the comment on the risk of disclosure of identity from some comments,  further details are below.

We would appreciate receiving your revised manuscript by Oct 24 2019 11:59PM. To enhance the reproducibility of your results, we recommend that if applicable you deposit your laboratory protocols in protocols.io, where a protocol can be assigned its own identifier (DOI) such that it can be cited independently in the future. For instructions see: http://journals.plos.org/plosone/s/submission-guidelines#loc-laboratory-protocols

We look forward to receiving your revised manuscript.

Kind regards,

Kathleen Finlayson

Academic Editor

PLOS ONE

Journal Requirements:

1. Please include a copy of the interview guide used in the study, in both the original language and English, as Supporting Information, or include a citation if it has been published previously.

2. Please amend either the title on the online submission form (via Edit Submission) or the title in the manuscript so that they are identical.

3. Please amend your manuscript to include your abstract after the title page.

Additional Editor Comments (if provided):

Thank you for submitting your manuscript for review. This is an interesting topic, and as such, I encourage you to consider the extensive feedback provided from the reviewers below.

Reviewers' comments:

Reviewer's Responses to Questions

Comments to the Author

1. Is the manuscript technically sound, and do the data support the conclusions?

Reviewer #1: Partly

Reviewer #2: Yes

2. Has the statistical analysis been performed appropriately and rigorously? 

Reviewer #1: Yes

Reviewer #2: Yes

3. Have the authors made all data underlying the findings in their manuscript fully available?

Reviewer #1: Yes

Reviewer #2: Yes

4. Is the manuscript presented in an intelligible fashion and written in standard English?

Reviewer #1: Yes

Reviewer #2: Yes

5. Review Comments to the Author

Reviewer #1: This manuscript describes interviews with individuals involved with clinical trial recruitment - the planning and design phase. The rationale provided is a bit weak. It does not reference recruitment models. The methods are adequately described; missing is the typical length of the interview. Results are presented with sufficient supportive quotes provided. The discussion is wide ranging and includes areas not addressed within the interviews/data/results. It sounds a bit 'preachy' in places and reads as if the authors had conclusions in mind (e.g. use of evidence to inform selected strategies; its time we create a culture change). The discussion includes items that might better be presented in results and could be significantly shortened.

A typo in the Results section: The encompass should be They encompass

Reviewer #2: This manuscript presents the results of a qualitative study of professionals who run clinical trial recruitment efforts. There were 23 interviews conducted. The interviews were semi-structured, in-depth interviews. The results were analyzed using the Framework method. Topics addresssed include: who is responsible for planning recruitment, detailed planning procedures, using empirical evidence, barriers to planning, and stress on those responsible for recruitment. Each topic is addressed with quotations from those who were interviewed. In general, a lack of resources (especially time) make it difficult for those planning recruitment to have sufficiently detailed plans to respond to issues in recruitment. Recruitment often lags behind expected results. Recruiters respond with ad hoc solutions, but the work is stressful as a result.

This paper presents interesting results. It raises important questions about how clinicial trials are typically run. I am a survey methodologist by training. As such, it is hard for me to evaluate this paper with respect to other papers on a similar topic. However, as a survey methodologist, I see a number of points of overlap with concerns in our field. There are also some differences. Expanding the paper in several areas would reinforce the importance of the results.

Major Comments

1. As with recruitment to clinical trials, surveys are facing rising nonresponse rates. This has been true since the 1990s and an area of concern. This manuscript argues that the major threat is to the power of studies. Clinical trials with recruitment difficulties might not meet their recruitment targets and, therefore, would be underpowered for drawing conclusions about treatment effectiveness. While this certainly is a risk, in the survey context, we have become more concerned with how this difficulty might lead to bias in estimates. This manuscript doesn't really discuss the potential for bias. It would be useful to review the pertinent literature on this question for clinical trials. I found this citation:

Hofer, A., M. Hummer, R. Huber, M. Kurz, T. Walch and W. W. Fleischhacker (2000). "Selection Bias in Clinical Trials with Antipsychotics." Journal of Clinical Psychopharmacology 20(6): 699-702.

As I am not an expert on the risk of bias in recruitment to clinical trials, I would like to hear more about this from a clinical trial perspective.

This raised a larger question that the field of survey methodology has been confronting for several years: what is quality in this context? The authors seem to suggest that meeting targets is the sole measure of quality. Are there other relevant measures? In survey methodology, we have developed measures related to the risk of nonresponse bias, but clinical trials might require other measures. This issue may be larger than this paper.

2. A second area of overlap between the two fields is the problem of uncertainty in the design. In surveys, the ability to predict how survey production will go has been greatly hampered by the general trend of declining response rates. What are the reasons for the uncertainty in recruitment to clinical trials? The paper quotes one of the interviewees as saying "the number of cases promised in any clinical study must be divided by a factor of at least 10." Is this due to a changing environment or is it because the designs in proposals to funders are overly optimistic? Some description of the sources of this uncertainty would be helpful.

In surveys, a technique known as responsive survey design was developed that planned for uncertainty. It's akin to risk management. Design changes are pre-planned and then triggered when indicators (such as response rates) meet (or don't meet) specified targets. I provide a key citation:

Groves, R. M. and S. G. Heeringa (2006). "Responsive design for household surveys: tools for actively controlling survey errors and costs." Journal of the Royal Statistical Society: Series A (Statistics in Society) 169(3): 439-457.

3. As an outsider, it seems that an important question for this field is how to systematize a body of knowledge. Should there be more academic research in the area of methods of recruitment to clinical trials? Could new journals be organized? Is there sufficient professional training available? Do people working in this area have conferences where they can share their knowledge? All of these would be useful and might help stimulate more systematic research on this topic and adoption of new techniques in practice. The conclusion of the paper makes some very general suggestions. "Trial Forge" is mentioned as one program aimed at improving design and implementation of studies. It would be helpful to make some more specific recommendations.

Survey methodology does have a fair amount of published research on recruitment methods and data quality. This research is clustered in a few journals that are associated with the profession (e.g. Public Opinion Quarterly). The role of questionnaire length is one such topic. The use of incentives is the subject of a huge amount of research. Some of this might be helpful for persons running recruitment to clinical trials.

4. General comment on disclosure risk. You have a relatively small sample. I worry that the information in Table 1, plus the comments with descriptive tag lines, could lead to identification of some of the participants. I can't say for sure, but wonder if someone in this community could identify colleagues and determine which quotes were theirs. For example, in Table 1, I can see that there is one person from a particular location. With that knowledge, and knowing who works there, could I figure out who some of the quotes were from?

Minor Comments

1.Sample size. I'm not convinced by a concept of "data saturation" stated in such a general way. I think the sample size should be a function of the thing being studied. For this research, I'm not very worried about justifying the sample size.

2.Organization of paper. The heads and subheads are confusing. It's difficult to see which sections are embedded in which.

6. PLOS authors have the option to publish the peer review history of their article (what does this mean?). If published, this will include your full peer review and any attached files.

Do you want your identity to be public for this peer review? For information about this choice, including consent withdrawal, please see our Privacy Policy.

Reviewer #1: No

Reviewer #2: Yes: James Wagner

---

## [Author Response · Author response to Decision Letter 0]

17 Oct 2019

Title: Using evidence when planning for trial recruitment: An international perspective from time-poor trialists

Authors: Heidi Gardner, Shaun Treweek, Katie Gillies

Manuscript ID: PONE-D-19-17368

11th October 2019

Dear Editor, 

Thank you very much for considering our manuscript for publication in PLOS One. We appreciate the reviewer’s considered responses and are pleased to have been able to strengthen the manuscript by incorporating their points into our revised submission.

The list below addresses each point raised by the reviewers in turn and identifies where new information to address the comment is included in the revised article. We have provided an amended manuscript with the changes tracked. 

I look forward to hearing from you. 

Best wishes, 

Heidi Gardner

 

Editor comments:

1. Please include a copy of the interview guide used in the study, in both the original language and English, as Supporting Information, or include a citation if it has been published previously.

We have added in interview guides used for both stakeholder groups (Recruiters and Designers), as S1 and S2 files. 

2. Please amend either the title on the online submission form (via Edit Submission) or the title in the manuscript so that they are identical.

Thank you for highlighting this. We have changed the title on the manuscript to ensure that it matches up with the online submission form. 

3. Please amend your manuscript to include your abstract after the title page.

Amended as requested.

Reviewer comments:

Reviewer #1: This manuscript describes interviews with individuals involved with clinical trial recruitment - the planning and design phase. The rationale provided is a bit weak. It does not reference recruitment models. 

With this project we aimed to explore if and how evidence is used in the process of recruitment planning by those tasked with designing or implementing recruitment strategies. We did not want to impose models on participants but rather see what they themselves raised. As it was, participants did not highlight specific recruitment models in their interviews and we therefore do not draw attention to details of recruitment models in the introduction and rationale of the manuscript. In addition, there are very limited numbers of robustly-evaluated strategies to support participant recruitment (ST and HG lead two systematic reviews of recruitment strategy evaluations), and we made the conscious decision not to draw attention to models and strategies that are not backed by robust evidence. As we found in our interviews, even where there is evidence to support a recruitment approach, our interviewees did not look for it. 

The methods are adequately described; missing is the typical length of the interview. 

We have provided details regarding interview length at the beginning of the Results section, “Twenty-five trialists from the Recruiter and Designer stakeholder groups were invited to secure 23 interviews which lasted between 32 and 77 minutes (median: 58 minutes).” We believe this is a sufficient level of detail for the manuscript.

Results are presented with sufficient supportive quotes provided. 

The discussion is wide ranging and includes areas not addressed within the interviews/data/results. It sounds a bit 'preachy' in places and reads as if the authors had conclusions in mind (e.g. use of evidence to inform selected strategies; it’s time we create a culture change). The discussion includes items that might better be presented in results and could be significantly shortened.

We have shortened the Discussion as suggested from 7 pages double-spaced to just under 6. Half a page of this is a response to Reviewer 2 comment #1. We’ve also removed some of the parts that could be called preachy too, and avoided duplicating presentation of things that already appear in Results.

A typo in the Results section: The encompass should be They encompass

Amended as requested.

Reviewer #2: 

Major Comments

1. As with recruitment to clinical trials, surveys are facing rising nonresponse rates. This has been true since the 1990s and an area of concern. This manuscript argues that the major threat is to the power of studies. Clinical trials with recruitment difficulties might not meet their recruitment targets and, therefore, would be underpowered for drawing conclusions about treatment effectiveness. While this certainly is a risk, in the survey context, we have become more concerned with how this difficulty might lead to bias in estimates. This manuscript doesn't really discuss the potential for bias. It would be useful to review the pertinent literature on this question for clinical trials. I found this citation:

Hofer, A., M. Hummer, R. Huber, M. Kurz, T. Walch and W. W. Fleischhacker (2000). "Selection Bias in Clinical Trials with Antipsychotics." Journal of Clinical Psychopharmacology 20(6): 699-702.

As I am not an expert on the risk of bias in recruitment to clinical trials, I would like to hear more about this from a clinical trial perspective.

The reviewer raises an interesting and important point. This is not strictly within the scope of our project, but we have added a section on ‘Looking beyond improved recruitment rates’ to the Discussion section to ensure that the reader is aware of this issue:

This project aimed to provide researchers with evidence about if and how the recruitment process is planned. We hope that this information, along with a significant body of additional work that has been published, planned and conducted, can be used to ultimately improve trial recruitment figures. We focus on the issue of low participant recruitment not because we believe it will solve every issue that a clinical trial may have, but because it is an avoidable issue that too many trials encounter. 

The potential for selection/participant bias [49] is a problem that we need to be aware of when considering the participants that we recruit [50], and whether they are retained until the end of the trial. Planning inclusive recruitment processes that provide the general population of patients that the trial’s results may apply to with opportunities to participate, is imperative to ensuring that trials provide useful results.

This raised a larger question that the field of survey methodology has been confronting for several years: what is quality in this context? The authors seem to suggest that meeting targets is the sole measure of quality. Are there other relevant measures? In survey methodology, we have developed measures related to the risk of nonresponse bias, but clinical trials might require other measures. This issue may be larger than this paper.

We agree with the issue raised by the reviewer – there is certainly a larger question regarding the concept of quality, and how it can be measured. As the reviewer highlights, this issue is larger than this paper, but we have added the ‘Looking beyond improved recruitment rates’ part of the Discussion section which highlights the point raised in the comment above and provides a comment on applicability of trial results. 

2. A second area of overlap between the two fields is the problem of uncertainty in the design. In surveys, the ability to predict how survey production will go has been greatly hampered by the general trend of declining response rates. What are the reasons for the uncertainty in recruitment to clinical trials? The paper quotes one of the interviewees as saying, "the number of cases promised in any clinical study must be divided by a factor of at least 10." Is this due to a changing environment or is it because the designs in proposals to funders are overly optimistic? Some description of the sources of this uncertainty would be helpful.

In surveys, a technique known as responsive survey design was developed that planned for uncertainty. It's akin to risk management. Design changes are pre-planned and then triggered when indicators (such as response rates) meet (or don't meet) specified targets. I provide a key citation:

Groves, R. M. and S. G. Heeringa (2006). "Responsive design for household surveys: tools for actively controlling survey errors and costs." Journal of the Royal Statistical Society: Series A (Statistics in Society) 169(3): 439-457.

The quote that the reviewer highlights is from Muench’s Third Law which is referenced within the text. That section (Managing Expectations within the Discussion) then goes on to describe why these overly optimistic figures are so often used when forecasting recruitment rates – inappropriately anchoring estimates by focussing on positive past experiences, failing to consider significant differences between studies, and the ‘Lake Wobegon Effect’ where individuals overestimate their achievements relative to the average.

3. As an outsider, it seems that an important question for this field is how to systematize a body of knowledge. Should there be more academic research in the area of methods of recruitment to clinical trials? Could new journals be organized? Is there sufficient professional training available? Do people working in this area have conferences where they can share their knowledge? All of these would be useful and might help stimulate more systematic research on this topic and adoption of new techniques in practice. The conclusion of the paper makes some very general suggestions. "Trial Forge" is mentioned as one program aimed at improving design and implementation of studies. It would be helpful to make some more specific recommendations.

Hearing the perspective of this reviewer, “an outsider”, is incredibly useful here. There are clusters of journals and conferences that we have not mentioned as we incorrectly assumed that readers would know about them. We have now added specific details of these resources to the Conclusion section in order to strengthen our recommendations:

This study has highlighted the complexity of planning trial recruitment strategies. The work involved can be lengthy and is often rushed as a result of time pressure. It is important that trialists, regulators, and funders recognise this process as an essential part of a trial’s workload, and that as a community we seek to alleviate the barriers and enhance the facilitators to effective recruitment planning. When trialists experience poor recruitment, they tend to implement multiple strategies based on experiential evidence; meaning that robust empirical evidence is rarely generated. The problem is not that there is nowhere for manuscripts covering these kinds of topics to be published (Trials, PLOS One, BMJ Open, and the Journal of Clinical Epidemiology have all published trials methodology research), the problem is that trialists are time-poor and therefore struggle to find the time to test the methods that they are implementing. Where robust evidence does exist, we must all work to ensure that trialists have unobstructed access to, know about and use these rigorously evaluated strategies that have been rigorously evaluated and proven to improve recruitment figures. Often these strategies are shared at relevant conferences such as the biennial International Clinical Trials Methodology Conference (UK-based) and the annual Society of Clinical Trials Meeting, but as these events require in person attendance, dissemination and expertise sharing is limited to those that can attend. Video recording or remote attendance using online conferencing suites would be one way to improve the accessibility of these events, therefore facilitating knowledge exchange and sharing of expertise with trialists around the world.

Survey methodology does have a fair amount of published research on recruitment methods and data quality. This research is clustered in a few journals that are associated with the profession (e.g. Public Opinion Quarterly). The role of questionnaire length is one such topic. The use of incentives is the subject of a huge amount of research. Some of this might be helpful for persons running recruitment to clinical trials.

As above, we agree that this needs to be make clear. We have added the following text to the Conclusion section:

If and when evidence-based approaches have been exhausted, we need to encourage implementation of embedded studies that effectively generate evidence that is useful to the wider trials community. There is a significant body of literature on survey methodology covering topics such as incentives and questionnaire length/content, which may act as a source of inspiration for testing strategies that have not yet been tested in the trial recruitment sphere. In addition, the Northern Ireland Hub for Trials Methodology Research holds the SWAT repository [52], which provides details (including outcome measures and analysis plans) of ongoing SWATs that can be implemented by trial teams. Once multiple trials have tested the same SWAT, the results can then be pooled to generate results that shed light on if and how the intervention operates across trials in a variety of contexts.

4. General comment on disclosure risk. You have a relatively small sample. I worry that the information in Table 1, plus the comments with descriptive tag lines, could lead to identification of some of the participants. I can't say for sure, but wonder if someone in this community could identify colleagues and determine which quotes were theirs. For example, in Table 1, I can see that there is one person from a particular location. With that knowledge, and knowing who works there, could I figure out who some of the quotes were from?

We appreciate the reviewer’s concern regarding potential for identification. During the consent process we explained how quotes may be used, and participants were given the opportunity for their quotes to be omitted from any publications or presentations resulting from this work. Only 4 of the 23 interviewees chose not to give consent for their anonymised quotes to be used, the remaining 19 participants were happy for their quotes to be used. We have adhered to their wishes. 

In addition, it is common for individuals in the trials community (particularly those in our Designer stakeholder group) hold multiple roles – e.g. Principal Investigator and Clinician – Participant 2, Professor and Chief Investigator – Participant 16, that, combined with our relatively small sample size would make it very difficult for someone in this community to identify colleagues. We have chosen to keep the descriptive tag lines as they are (role, stakeholder group, location, participant number), as each of these descriptors adds a layer of context to the quote being presented, taking out location details for example, would limit that context and reduce value for the reader. 

Minor Comments

1.Sample size. I'm not convinced by a concept of "data saturation" stated in such a general way. I think the sample size should be a function of the thing being studied. For this research, I'm not very worried about justifying the sample size.

We agree. We have chosen to keep the ‘Sample size’ section in the manuscript for the benefit of readers that may be less familiar with qualitative research and/or may disagree with this perspective. 

2.Organization of paper. The heads and subheads are confusing. It's difficult to see which sections are embedded in which.

We have amended the headings and subheadings to reflect PLOS guidelines.

---

## [Editor Report · Decision Letter 1]

20 Nov 2019

Using evidence when planning for trial recruitment: An international perspective from time-poor trialists

PONE-D-19-17368R1

Dear Dr. Gardner,

We are pleased to inform you that your manuscript has been judged scientifically suitable for publication and will be formally accepted for publication once it complies with all outstanding technical requirements.

With kind regards,

Kathleen Finlayson

Academic Editor

PLOS ONE
---

## [Editor Report · Acceptance letter]

27 Nov 2019

PONE-D-19-17368R1 

­­­­Using evidence when planning for trial recruitment: An international perspective from time-poor trialists

Dear Dr. Gardner:

I am pleased to inform you that your manuscript has been deemed suitable for publication in PLOS ONE. Congratulations! Your manuscript is now with our production department. 

With kind regards,

on behalf of

Dr. Kathleen Finlayson 

Academic Editor

PLOS ONE